# A Pharmacokinetic and Pharmacodynamic Evaluation of the Anti-Hepatocellular Carcinoma Compound 4-*N*-Carbobenzoxy-gemcitabine (Cbz-dFdC)

**DOI:** 10.3390/molecules25092218

**Published:** 2020-05-08

**Authors:** Yilin Sun, Jiankun Wang, Kun Hao

**Affiliations:** Laboratory of Drug Metabolism and Pharmacokinetics, China Pharmaceutical University, 24 Tongjia Xiang, Nanjing 210009, China; cpulili2015@163.com (Y.S.); wangjiankun789@126.com (J.W.)

**Keywords:** 4-*N*-carbobenzoxy-gemcitabine, pharmacokinetic, pharmacodynamic, prodrug, carboxylesterase-1

## Abstract

Gemcitabine (dFdC) demonstrates significant effectiveness against solid tumors in vitro and in vivo; however, its clinical application is limited because it tends to easily undergo deamination metabolism. Therefore, we synthesized 4-*N*-carbobenzoxy-gemcitabine (Cbz-dFdC) as a lead prodrug and conducted a detailed pharmacokinetic, metabolic, and pharmacodynamic evaluation. After intragastric Cbz-dFdC administration, the C_max_ of Cbz-dFdC and dFdC was 451.1 ± 106.7 and 1656.3 ± 431.5 ng/mL, respectively. The T_max_ of Cbz-dFdC and dFdC was 2 and 4 h, respectively. After intragastric administration of Cbz-dFdC, this compound was mainly distributed in the intestine due to low carboxylesterase-1 (CES1) activity. Cbz-dFdC is activated by CES1 in both humans and rats. The enzyme kinetic curves were well fitted by the Michaelis–Menten equation in rats’ blood, plasma, and tissue homogenates and S9 of the liver and kidney, as well as human liver S9 and CES1 recombinase. The pharmacodynamic results showed that the Cbz-dFdC have a good antitumor effect in the HepG2 cell and in tumor-bearing mice, respectively. In general, Cbz-dFdC has good pharmaceutical characteristics and is therefore a good candidate for a potential prodrug.

## 1. Introduction

Gemcitabine (dFdC) is a deoxycytidine analog that demonstrates significant effectiveness against solid tumors in vitro and in vivo, through the inhibition of DNA synthesis by incorporation into nascent RNA and DNA strands, resulting in cytotoxicity [1,2,3,4,5,6]. It can be used for the treatment of various types of cancers, such as hepatocellular carcinoma, bladder cancer, and so on [7,8,9,10]. However, there are still some problems with its potential clinical application. After intravenous injection, dFdC can be metabolized via cytidine kinase into active monophosphate, diphosphate, and triphosphate metabolites, but these three active metabolites of dFdC are rare, accounting for only about 10% of the dose of dFdC [11,12,13,14,15]. Extensive deamination of dFdC takes place in plasma and the liver via cytidine deaminase, producing the inactive metabolite 2’,2’-difluorodeoxyuridine (dFdU, Figure 1) [11,12,16,17]. These problems lead to a short terminal half-life of dFdC, where an increasing dose not only increases the efficacy but also increases the toxicity.

The affinity between dFdC and cytidine kinase is higher than that between dFdC and cytidine deaminase, but cytidine kinase is easily saturated, leading to extensive deamination and the production of a high number of inactive metabolites of dFdU in vivo. However, prolonged sustained exposure by a low concentration of dFdC increases the proportion of active metabolites. In order to provide a flat and long-term dFdC concentration in plasma and tissue, different 4-(*N*)-carbamate-modified dFdC prodrugs that utilize the high cytidine deaminase stability of the carbamate portion, such as LY2334737 (Figure 1), squalenoyl-dFdC, stearyl-dFdC, and so on, have been developed to extend the systemic exposure of dFdC [18,19,20,21]. These developments prompted us to further explore whether other carbamate modifications can also be applied to optimize the pharmacokinetic properties of dFdC [22,23,24,25].

In this study, we synthesized 4-(*N*)-carbobenzoxy-gemcitabine (Cbz-dFdC) (Figure 1) and established a UFLC-MS/MS detection method [26] to simultaneously quantify Cbz-dFdC and dFdC, for further pharmacokinetic and pharmacodynamic evaluation. Although both LY2334737 and Cbz-dFdC are 4-(*N*)-carbamate-substituted derivatives of dFdC, they are different in terms of the substituent types. LY2334737 is an acyl-modified dFdC derivative, while Cbz-dFdc is an alkoxycarbonyl-modified dFdC derivative. The acyl group in LY2334737 is more sterically hindered, which slows the hydrolysis of the prodrug and impedes a release of dFdC in vivo. The released stability of alkoxycarbonyl moieties is often better than that of amide structures. When alkoxycarbonyl moieties are applied to the prodrug design, the design is likely to exhibit more ideal pharmacokinetic properties and achieve a slow release of dFdC in the body. The kinds of alkoxycarbonyl-modified dFdC derivatives have not been studied, so it is worth trying to study the Cbz-dFdC properties. Carboxylesterase (CES) is one of the serine esterase enzymes found in many animals and in humans [27,28]. Based on sequence homology, Satoh and Hosokawa proposed classifying CES isoforms into four families. CES1, CES2, CES3, and CES4 exist in animals, and CES1 and CES2 exist in humans [29]. CES1 is one of the most important hydrolytic enzymes in humans and animals. Its hydrolysis comprises over 80% of the total hydrolytic activity [30]. CES2 is another important hydrolytic enzyme involved in the hydrolysis metabolism. CES1 and CES2 mediate the hydrolysis of various endogenous and exogenous substances in vivo and also play an important role in the metabolic activation process of many prodrugs, such as oseltamivir and angiotensin-converting enzyme inhibitors [31,32]. As a alkoxycarbonyl-modified dFdC prodrug, we hypothesized that Cbz-dFdC is also mainly hydrolyzed by CES, so we investigated the key enzyme kinetics for Cbz-dFdC hydrolysis in this study, in addition to the pharmacokinetics and pharmacodynamics of Cbz-dFdC.

## 2. Results

### 2.1. Plasma Pharmacokinetics In Vivo

After administering Cbz-dFdC, the plasma concentrations of Cbz-dFdC and dFdC were detected and were calculated from the calibration curves. The average concentration–time curve of the two substances is shown in Figure 2A. After intragastric administration of Cbz-dFdC, the C_max_ value of Cbz-dFdC and dFdC was 451.1 ± 106.7 and 1656.3 ± 431.5 ng/mL, respectively, and the corresponding mean T_max_ value was 2.00 ± 0.00 and 4.00 ± 0.00 h. Compared with the CBZ-dFdC concentration, the plasma concentrations of dFdC were significantly higher after intragastric Cbz-dFdC administration, while the T_max_ values were significantly delayed to 4 h. The plasma concentration of dFdC remained about 300 ng/mL at 24 h, and the mean AUC_0–∞_ was 27,498.0 ng·h/mL. After intravenous injection of dFdC at 15 mg/kg (equimolar dose to Cbz-dFdC), dFdC was rapidly eliminated from the body (Figure 2B). The initial plasma concentration of dFdC was high (about 15,000 ng/mL), but it decreased rapidly to about 130 ng/mL at 24 h, the mean AUC_0–∞_ was 22,685 ng·h/mL. The non-compartmental pharmacokinetic parameters are shown in Table 1.

### 2.2. Tissue Distribution

After intragastric administration of Cbz-dFdC (22 mg/kg) and intravenous injection of dFdC (15 mg/kg) in the rats, the CES1 activity and drug concentrations of Cbz-dFdC and dFdC in the liver, kidney, and intestine were determined. The tissue concentrations of Cbz-dFdC and dFdC in different tissues after intragastric administration of Cbz-dFdC (22 mg/kg) are schematically shown in Figure 3A,B. The results demonstrated that Cbz-dFdC and dFdC were distributed in the liver, kidney, and intestine. The Cbz-dFdC level in the intestines was significantly higher than that in the other tissues, especially at 0.5 h; correspondingly, the concentration ratio of Cbz-dFdC to dFdC was significantly lower in the kidney and liver, than in the intestine, suggesting that Cbz-dFdC in the intestine was not easily degraded. The histogram of enzyme activity in the different tissues showed that the intestinal CES1 activity was significantly lower than that of the kidney and liver, suggesting that CES1 might mediate the degradation of Cbz-dFdC and produce the high concentration of Cbz-dFdC in the intestine (Figure 3C). After intravenous injection of dFdC (15 mg/kg) in the rats, the tissue concentrations of dFdC in different tissues are schematically shown in Figure 3D.

### 2.3. Enzymatic Kinetics In Vitro

In order to verify whether Cbz-dFdC was catalyzed by hydrolysis to dFdC by CES1, we used rats’ blood, plasma, and tissue homogenates and S9 of the liver, kidney, and intestine as well as human plasma and S9 of the human liver, kidney, and intestine for incubation in vitro. The enzyme kinetic curve based on the disappearance of Cbz-dFdC and appearance of dFdC in rat tissues is shown in Figure 4. In rat blood, plasma, and tissue homogenates and S9 of the liver and kidney, Cbz-dFdC was easily degraded, and dFdC was generated. The Cbz-dFdC in rat intestine homogenate and S9 was essentially non-degradable, and dFdC were not generated. The enzyme kinetic curve and parameters could not be graphed and estimated due to non-degradable Cbz-dFdC in rat intestine homogenate and S9. It has been reported that CES1 shows tissue specificity in rats, where CES1 is highly expressed in the liver, kidney, and plasma. CES2 is highly expressed in the intestine, but not in the liver, kidney, and plasma. Therefore, it can be concluded that Cbz-dFdC is easily metabolized into dFdC by CES1 in rats but is relatively stable in CES2. The enzymatic kinetic parameters of Cbz-dFdC in rats’ blood, plasma, and homogenates and S9 of the liver and kidney are shown in Table 2.

As shown in Figure 5, it could be concluded that Cbz-dFdC is unstable in human liver S9 and is easily hydrolyzed into dFdC. In human plasma and S9 of the kidney and intestine, Cbz-dFdC did not degrade, where the enzyme kinetic curve and parameters could not be graphed and estimated in human plasma and in S9 of the kidney and intestine. CES1 is mainly expressed in the human liver but has almost no expression in the intestine. CES2 is highly expressed in the intestine but rarely in the liver of humans. In addition, CES1 expression in the kidney is very low in humans, and about 95% of the renal hydrolytic activity is derived from CES2 [33,34,35]. Based on the above, it can also be concluded that Cbz-dFdC is easily hydrolyzed by CES1 in humans and is relatively stable in CES2. The enzymatic kinetic parameters of S9 of the human liver and CES1 recombinase are also shown in Table 2.

### 2.4. Cell Efficacy and Unptake

The ability of Cbz-dFdC and dFdC to inhibit tumor cell growth in vitro was detected by the CCK-8 assay using the human HepG2 cell line. As shown in Figure 6A, the representative concentration–growth inhibition curve showed the inhibitory effect of Cbz-dFdC and dFdC in a concentration-dependent manner, which suggest that Cbz-dFdC has a certain cytotoxicity. It is true that the anti-tumor effect of dFdC in HepG2 cells was better than that of Cbz-dFdC. However, considering that dFdC was released slowly after intragastric administration of Cbz-dFdC, Cbz-dFdC might be supposed to have some advantages in terms of an anti-tumor effect in vivo.

We measured the cell uptake of Cbz-dFdC and dFdC in HepG2 cells Figure 6B. The results showed that—(1) after addition of Cbz-dFdC, the concentration of Cbz-dFdC in HepG2 cells was low, and then hydrolyzed to dFdC in different concentrations; (2) after the dFdC treatment alone, the intracellular concentrations of dFdC in HepG2 cells also increased with the increase of the dFdC concentration; (3) the intracellular concentration of dFdC after the dFdC treatment alone was higher than that of dFdC, after the Cbz-dFdC treatment. These results might be one of the reasons that the anti-tumor effect of dFdC in HepG2 cells was better than that of Cbz-dFdC.

### 2.5. Antitumor Efficacy In Vivo

We compared the anti-tumor activity between dFdC and Cbz-dFdC in mice (In Figure 7). At equimolar dose of dFdC (15 mg/kg) and Cbz-dFdC (22 mg/kg), dFdC was treated by i.p. 15 mg/kg/day, while Cbz-dFdC was treated by i.g. 23 mg/kg/day. The results showed that the antitumor effects of Cbz-dFdC were slightly better than that of dFdC in vivo.

## 3. Discussion

In the present study, the 4-(*N*)-carbamate derivative Cbz-dFdC was synthesized. On the basis of the different activities of two hydroxyl groups and an amino group on dFdC, TMSCl was first used to protect the two hydroxyl groups, which were more active and generated intermediate products, 2’,5’-trimethylchlorosilane dFdC. Then, under the condition of no purification, benzyl chloroformate was added directly for the acylation of amidogen, and benzyl chloroformate acylation TMSCl automatically fell off during the process, generating the final product Cbz-dFdC. 13C NMR (126 MHz, DMSO) δ 163.38 (C), 153.99 (C), 153.00 (C), 144.46 (CH), 135.81 (C), 128.48 (CH)2, 128.19 (CH), 127.96 (CH)2, 122.93 (t, JC − F = 259.6, C), 94.86 (CH), 84.08 (t, JC − F = 30.2, CH), 80.98 (CH), 68.37 (t, JC − F = 22.7, CH), 66.67 (CH_2_), 58.78 (CH_2_).1H NMR (500 MHz, DMSO) δ 10.98 (s, 1H), 8.23 (d, *J.* = 7.6, 1H), 7.45–7.32 (m, 5H), 7.11 (d, *J.* = 7.6, 1H), 6.31 (d, *J.* = 6.5, 1H), 6.16 (t, *J.* = 7.4, 1H), 5.29 (t, *J.* = 5.5, 1H), 5.20 (s, 2H), 4.25–4.13 (m, 1H), 3.88 (dt, *J.* = 8.5, 3.1, 1H), 3.84 3.62 (m, 2H). MS (ESI) *m*/*z* 398.4 [M + H]^+^.

In our study, the cytotoxicity of Cbz-dFdC in vitro was lower than that of an equimolar dose of dFdC, which was speculated to be caused by the following reasons. First, when dFdC was made into derivatives, it was not easy to activate it to generate the active form. Second, due to there being less CES1 for intracellular reactions in vitro, Cbz-dFdC existed in a prototype form for a long time. However, Cbz-dFdC could effectively improve the pharmacokinetic properties of dFdC and played a superior role in the antitumor activity in vivo. It is well-known that dFdC is an intravenous formulation because dFdC has serious toxic effects on the gastrointestinal tract [36], which limits its clinical application and patient compliance. In addition, dFdC is a metabolically unstable drug and has a half-life of only 0.2 h and 2.0 h in humans and rats after intravenous injection, respectively [36]. Its poor compliance and metabolic instability have created a clinical need for an oral dFdC prodrug. In order to improve the pharmacokinetic properties of dFdC, we designed and synthesized the carbamate derivative Cbz-dFdC and evaluated its pharmacokinetics and pharmacodynamics in vivo and in vitro.

Based on our study results, it could be concluded that Cbz-dFdC is a good potential candidate for such a prodrug. Similar to previous pharmacokinetic study of intravenous injection of dFdC [37,38], the mean AUC of dFdC metabolized from Cbz-dFdC after intragastric Cbz-dFdC was significantly larger than that of dFdC via intravenous injection, in our pharmacokinetic experiment. Although the C_max_ of dFdC metabolized from Cbz-dFdC was significantly reduced, the T_max_ was significantly delayed, indicating that Cbz-dFdC plays a certain role in the slow release and long residence of dFdC in vivo. The plasma concentration of dFdC metabolized from Cbz-dFdC remained about 600 ng/mL at 12 h and 130 ng/mL at 48 h after intragastric Cbz-dFdC administration, but only about 200 ng/mL at 12 h and 40 ng/mL at 48 h after dFdC intravenous injection. These results showed that Cbz-dFdC plays a superior role in antitumor activity in vivo.

The tissue distribution in rats showed the following: (1) The Cbz-dFdC level in the intestine was significantly higher than that in other tissues, which suggests that Cbz-dFdC in the intestine is not easily degraded. (2) Compared to plasma concentration, the drug concentration in tissue was lower, suggesting a weak tissue penetration and slow entry of Cbz-dFdC and dFdC into tissues. (3) The histogram of enzyme activity in different tissues shows that the intestinal CES1 activity was significantly lower than that of the kidney and liver, suggesting that CES1 might mediate the degradation of Cbz-dFdC. Capecitabine is also a carbamate derivative of 5-FU, and it has been reported that the carbamate group is preliminarily activated by CES1 in vivo [22,39].

CES is a serine esterase enzyme, which is widely found in animals and humans [27,28] and is located in the smooth endoplasmic reticulum of tissue cells [40,41,42]. Based on sequence homology [29], two CES subtypes have been found to exist in humans, and four CES subtypes have been found to exist in rats, related to the metabolism of exogenous and endogenous substances [30,34,40,43,44,45]. Among the CES subtypes, CES1 plays an important role in the metabolic activation or inactivation of endogenous and exogenous substances in humans and rats, involving a large number of esters, amides, thioesters, carbamates, and other structural types [34,40,45]. The present enzyme kinetic results confirmed that there is tissue specificity in rats and humans. In rats, CES1 is highly expressed in the kidney and blood but not in the intestines. CES2 is highly expressed in the intestine but not in the kidneys and blood. The same results also showed that CES1 hydrolysis mainly occurs in S9, so plasma and S9 should mainly be used in the study of human CES catalysis. In humans, CES1 is mainly expressed in the human liver but has almost no expression in the intestine. CES2 is highly expressed in the intestine and rarely in the liver; CES1 is not highly expressed in the kidney of humans [33,34,35].

## 4. Materials and Methods

### 4.1. Materials and Equipment

Analytical-grade reagents such as dFdC, trimethylchlorosilane (TMSCl), anhydrous pyridine, and benzyl chloroformate (Cbz-Cl) were supplied by Energy (Shanghai, China) at approximately 98% purity. Dulbecco’s Modified Eagle’s medium and fetal bovine serum were purchased from Gibico (Shanghai, China). Human recombinant CES1, human S9 of the liver and kidneys, and intestines were purchased from Ruid Liver Disease Research Co., Ltd. (Shanghai, China). A bicinchoninic acid (BCA) kit was purchased from Beyotime Biotechnology (Shanghai, China). A CES1 activity kit was purchased from Jingmei Biotechnology Co., Ltd. (Suzhou, China). CuSO_4_, NaHCO_3_, NaCl, MgSO_4_, dichloromethane, and methanol were supplied by Energy (Shanghai, China). A Thermo Fisher electron LED GmbHD-37520 Biofuge (Osterode, Lower-Saxony, Germany) was used for plasma extraction. A Vortex Genie 2 (Scientific Industries, Bohemia, NY, USA) was used for mixing. NMR spectra were recorded on an ACF* 300Q and 500Q Bruker spectrometer (Bruker, Billerica, MA, USA). An AB Sciex API 4000 TM LC–MS/MS (Applied Biosystems, Waltham, MA, USA) was used to detect the concentration of the Cbz-dFdC and dFdC.

### 4.2. Animals

Adult male Sprague–Dawley (SD) rats (200 ± 20 g) were purchased from Shanghai SIPPR-BK Laboratory Animal Co., Ltd. (Changsha, China) and were housed in a standard room with a 12/12 h light/dark cycle, where the temperature and humidity were 22 ± 2 °C and 55 ± 5%, respectively. Before the experiments were conducted, the rats were allowed one week to adapt to the laboratory conditions, and standard food was provided during this time. Animal welfare and experimental procedures were strictly performed in accordance with the Guidelines for Animal Experimentation of China Pharmaceutical University (Nanjing, China), and the protocol was approved by the Animal Ethics Committee of the institution (approval number 2019-09-003). Each group of animals was housed in a cage and fasted overnight (12 h), before the experiments. The solvent used for intragastric and injected administration was 0.9% normal saline.

### 4.3. The Synthesis of Cbz-dFdC

The dFdC, TMSCl, and Cbz-Cl were prepared through the protection of the free hydroxyl groups and the amidation of the amino groups on the dFdC molecule. The Cbz-dFdC was synthesized in a one-step process. The schematic illustration of the synthetic route is shown in Figure 8. In brief, dFdC (1.0 equivalent) was first sealed with argon and dissolved in dry pyridine (2 mL). After the dissolution of dFdC, the TMSCl (4.0 equivalent) was added to the above solution at 0 °C, and the reaction was performed at room temperature for 1 h. Cbz-Cl (1.2 equivalent) was added to the above mixture and stirred overnight in an argon environment at room temperature. The crude products were diluted with ethyl acetate, washed with 10% CuSO_4_, saturated with NaHCO_3_ and NaCl, and dried over MgSO_4_. The solvent was removed under a vacuum, and the product was further separated by chromatography on silica, using 7% methyl alcohol in dichloromethane as the eluent. The product was taken up in anhydrous ether and stirred for 2 h to remove hydrophobic impurities. The solvent was removed by suction filtration to obtain a pure white solid. NMR spectra of Cbz-dFdC were provided as Appendix A. The mass spectrogram of Cbz-dFdC was provided as Appendix A. The content of Cbz-dFdC was determined by HPLC-UV with 98% purity (Appendix A).

### 4.4. Pharmacokinetics In Vivo

SD rats (*n* = 6) were intragastrically administered Cbz-dFdC (22 mg/kg) and intravenously injected with dFdC (15 mg/kg), and 150 μL blood samples were collected from the retro-orbital route in the right eye at 0, 5, 10, 20, and 30 min and at 1, 2, 4, 6, 12, 24, and 48 h, using heparinized tubes. Without delay, the blood was directly centrifuged at 8000× *g* at 4 °C for 5 min. Then, the supernatant plasma was collected and immediately extracted using a previously established method [26] or was frozen at −80 °C until further analysis.

### 4.5. Tissue Distribution

In order to investigate the drug distribution in different tissues and the relationship between CES1 activity and tissue concentration, we conducted tissue distribution experiments. The SD rats were divided into six groups (*n* = 6 for each group). After a single intragastric administration of Cbz-dFdC (22 mg/kg) and a single injected administration of dFdC (15 mg/kg), the SD rats were sacrificed at 0.5 h, 1.5 h, and 4 h, and the liver, kidney, and intestine were collected. The liver was perfused with ice-cold normal saline and then removed, followed by ice-cold normal saline cleaning. The kidney and intestine were directly removed and then washed with ice-cold normal saline. All of the above operations were carried out on ice, and the removed tissue was stored in an ice bath. The liver, kidney, and intestine were cut and crushed, homogenized with an appropriate amount of phosphate buffer by a homogenizer, and then diluted 5 times with ice-cold PBS buffer (in an ice bath). The tissue homogenates were determined with a BCA kit for protein concentration, a CES1 kit for CES1 activity, and UFLC–MS/MS for drug tissue concentration after pretreatment; the pretreatment method was reported previously [26].

### 4.6. Enzymatic Kinetics In Vitro

The SD rats without any treatment were sacrificed after adapting to the environment. Blank blood, the liver, the kidney, and the intestine were collected and treated as described in Section 4.4. Part of the homogenate was removed for centrifugation for 20 min at 9000× *g* at 4 °C, and the supernatant was obtained as tissue S9. The tissue homogenate and S9 were diluted to 10 ng of protein/mL after the protein concentration was determined by the BCA kit (Abcam, Cambridge, MA, USA). Each matrix consisted of 45 uL of the matrix mixed with 5 uL of the drug dissolved in normal saline solution for about 3 min, followed by incubation at 37 °C in a water bath. Then, acetonitrile containing the internal standard was added to terminate the reaction. Human tissue S9 and human CES1 recombinase purchased from the Ruid Liver Disease Research Co., Ltd. (Shanghai, China) were incubated under the same conditions. An enzyme kinetic study was obtained from the matrix following incubation at 37 °C for 30 min with 5, 10, 20, 50, 100, or 200 μM of Cbz-dFdC. We estimated the parameters and simulated curves according to the Eadie–Hofstee method.

### 4.7. Antitumor Efficacy and Uptake In Vitro

The cells were cultured on a 96-well plate for 4 h for each well. Then, dimethyl sulfoxide solution containing 0.1–100 mM of Cbz-dFdC and dFdC was diluted 1000 times with the medium before administration. A total of 200 uL of medium containing Cbz-dFdC and dFdC was added per well (*n* = 6). The medium with drugs was discarded, and 10% CCK-8 solution was added. Then, the absorbance value was read at a wavelength of 450 nm after incubation at 37 °C, for 30 min. The drug concentration of Cbz-dFdC and dFdC in cell were determined by a previously established method [26].

### 4.8. Antitumor Efficacy In Vivo

A cell suspension (2 × 10^7^ cells/mL) of HepG2 with 0.2 mL was subcutaneously injected into the right armpits of the mice. After the hepatocellular carcinomas were inoculated, mice bearing an overt tumor with a size of 100–200 mm^3^ were randomized into the control and treatment groups. One group was injected (i.p.) with PBS as the control group, Cbz-dFdC groups were given with 22 mg/kg/day (i.g.), respectively, and the dFdC groups were given with 15 mg/kg/day (i.p.), respectively. The tumor sizes of all mice were monitored at 0 d, 3 d, 6 d, 9 d, 12 d, and 15 d after drug treatment. The tumor size was converted into tumor volume (V), which was calculated by the formula (L × W^2^ × 0.52) and presented as the mean ± standard deviation, where L and W are the length and width of the tumor, respectively. The tumors were measured with a Vernier caliper until day 15, when the PBS brine-treated mice began to die, then all rats were sacrificed via nitrogen inhalation.

### 4.9. Data Analysis

Data were expressed as means ± standard deviations. Data analyses were performed by one-way ANOVA with Tukey multiple comparison test.

## 5. Conclusions

Cbz-dFdC exhibited sustained release action of dFdC in plasma pharmacokinetics, with slow and stable exposure. In vivo and in vitro, Cbz-dFdC was mainly activated by CES1 to generate free dFdC in both human tissue and rats. In addition, Cbz-dFdC showed certain cytotoxicity in HepG2, which had a good effect on the HepG2 tumor-bearing mice. This study provides a reference for the synthesis of carbamate derivatives of dFdC in the future.

## Figures and Tables

**Figure 1 molecules-25-02218-f001:**
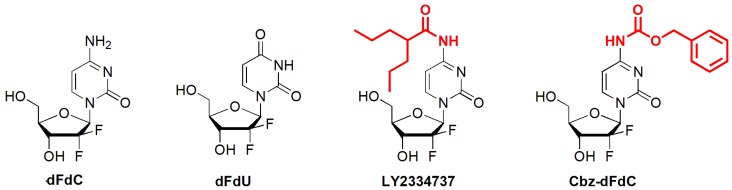
The structures of Cbz-dFdC; dFdC; dFdU; and LY2334737.

**Figure 2 molecules-25-02218-f002:**
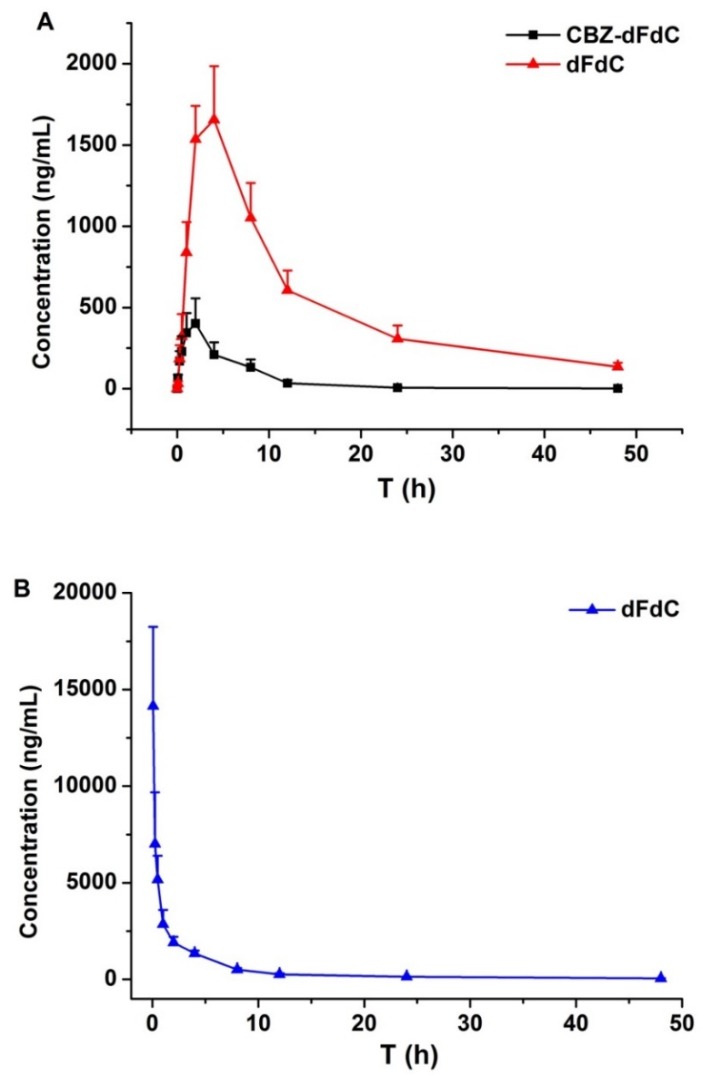
The plasma concentrations of Cbz-dFdC and dFdC in rats, (**A**) after i.g. administration of Cbz-dFdC (22 mg/kg); and (**B**) after i.v. administration of dFdC (15 mg/kg).

**Figure 3 molecules-25-02218-f003:**
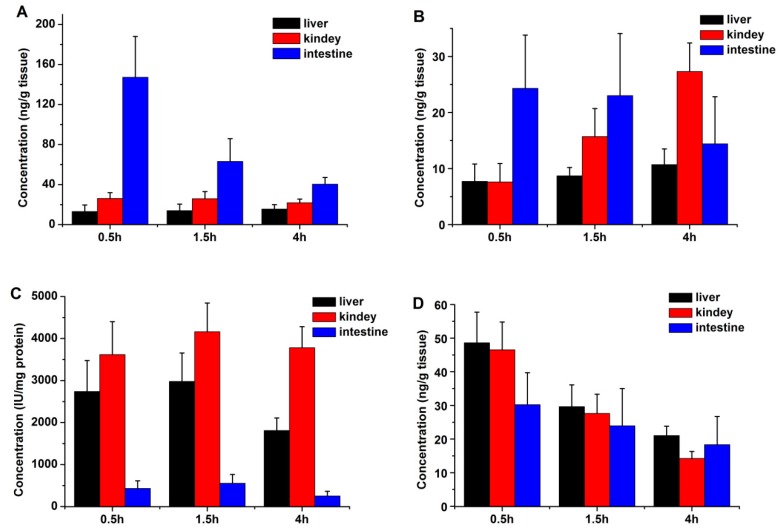
The tissue distributions of Cbz-dFdC and dFdC in rats. (**A**) Cbz-dFdC concentration after i.g. administration of Cbz-dFdC (22 mg/kg); (**B**) dFdC concentration after i.g. administration of Cbz-dFdC (22 mg/kg); (**C**) CES1 activity in different tissue; and (**D**) dFdC concentration after i.v. administration of dFdC (15 mg/kg).

**Figure 4 molecules-25-02218-f004:**
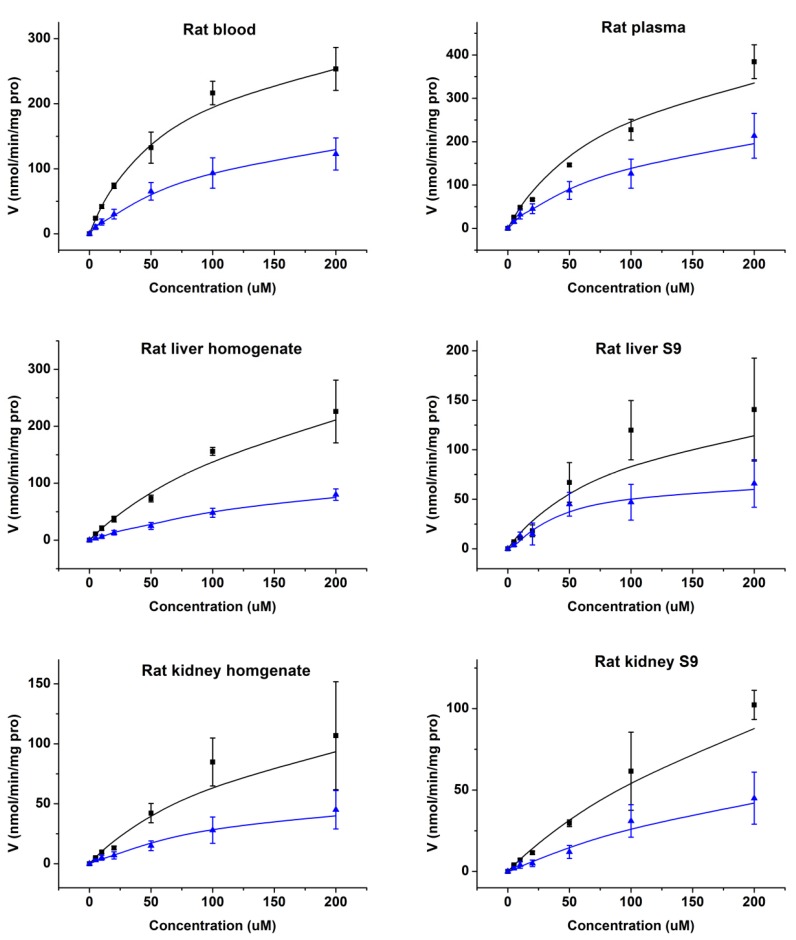
The enzyme kinetics of Cbz-dFdC in rat blood; plasma; tissue homogenate; and S9. Black square and black line represent the observed and simulated values of Cbz-dFdC disappearance in incubation. The blue triangle and blue line represent the observed and simulated values of the dFdC appearance in incubation.

**Figure 5 molecules-25-02218-f005:**
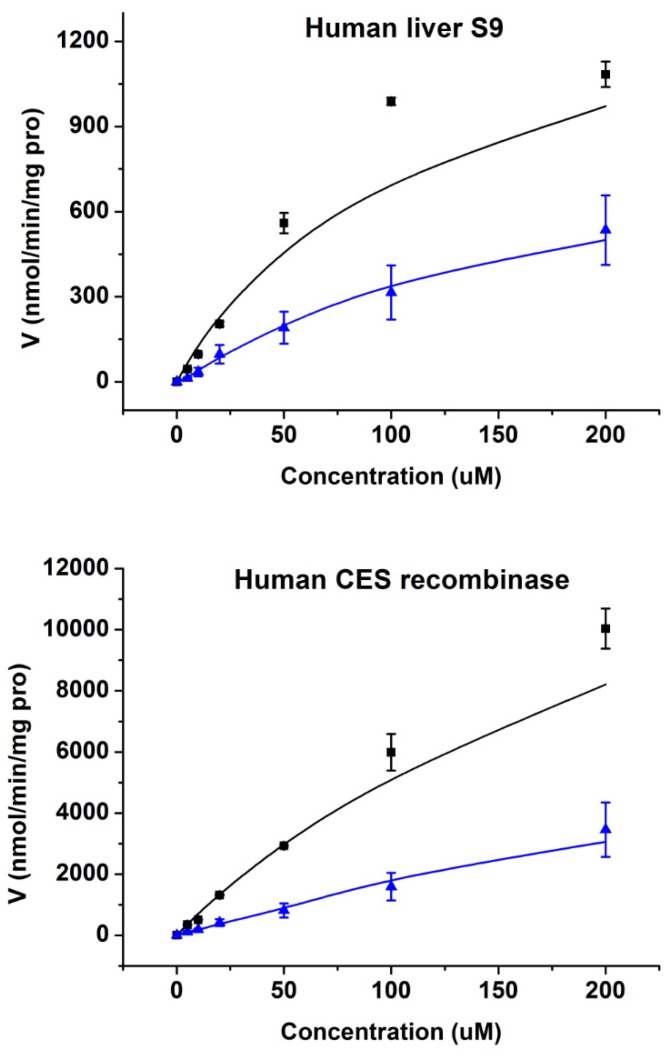
The enzyme kinetics of Cbz-dFdC in human liver S9 and CES1 recombinase. Black square and black line represent observed and simulated values of Cbz-dFdC disappearance in incubation. Blue triangle and blue line represent observed and simulated values of dFdC appearance in incubation.

**Figure 6 molecules-25-02218-f006:**
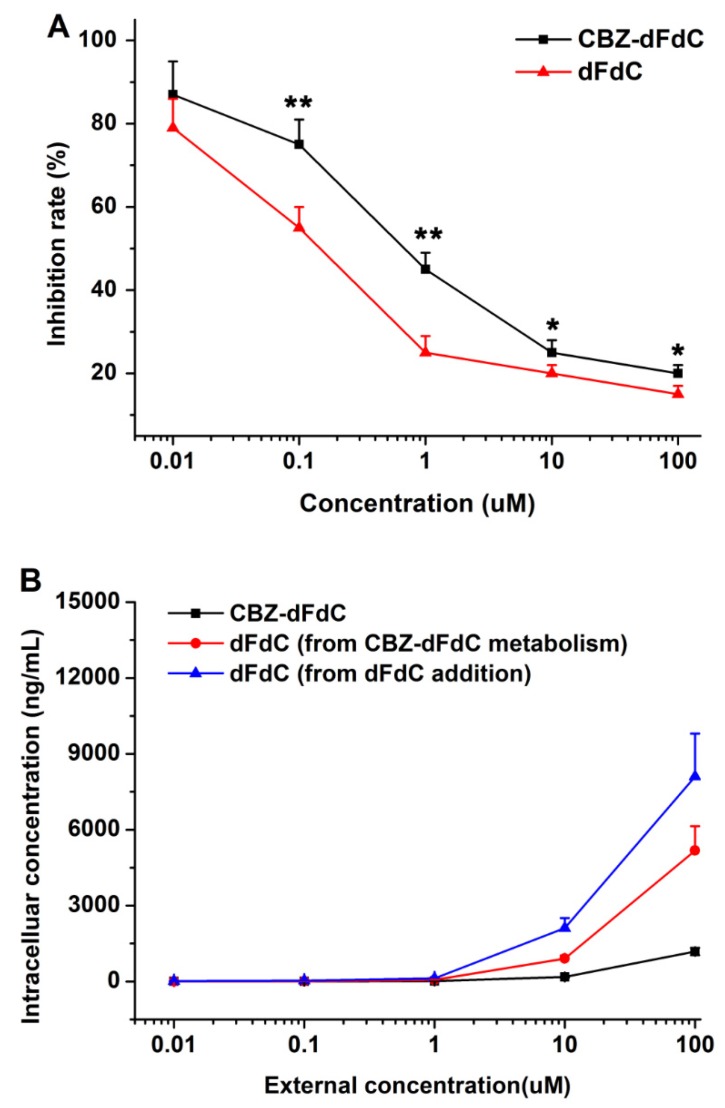
The cytotoxicity and uptake of Cbz-dFdC and dFdC in the HepG2 cell line. (**A**) The cytotoxicity of Cbz-dFdC and dFdC in the HepG2 cell line; and (**B**) the cell uptake of Cbz-dFdC and dFdC in the HepG2 cell line. Cbz-dFdC group were compared with dFdC group (* *p* < 0.05, ** *p* < 0.01).

**Figure 7 molecules-25-02218-f007:**
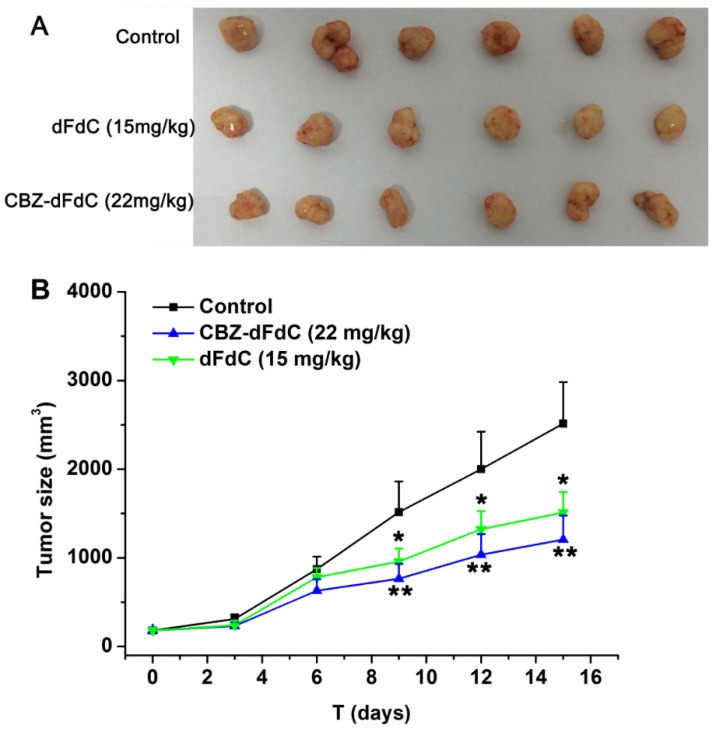
The measured anti-tumor effect after Cbz-dFdC administration (i.g., 22 mg/kg) and dFdC administration (i.p., 15 mg/kg) for 15 days, (**A**) tumor image; and (**B**) tumor size. The control group is represented by the solid black line. The dose of Cbz-dFdC (22 mg/kg) is represented by the blue line. The dose of dFdC (15 mg/kg) is represented by the green line. the Cbz-dFdC and dFdC groups were compared with the control group (* *p* < 0.05 and ** *p* < 0.01).

**Figure 8 molecules-25-02218-f008:**
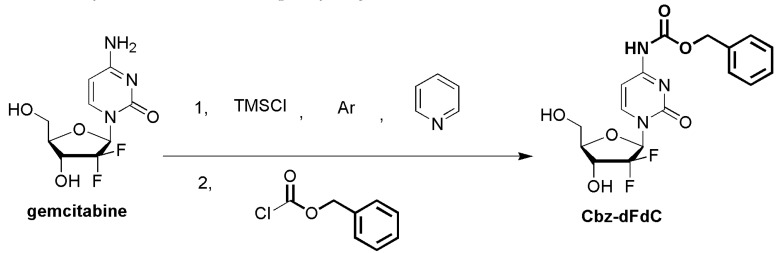
The synthetic route of Cbz-dFdC.

**Table 1 molecules-25-02218-t001:** Pharmacokinetic parameters of Cbz-dFdC and dFdC.

Parameters	Cbz-dFdC (i.g. 22 mg/kg)		dFdC (i.v. 15 mg/kg)
	Cbz-dFdC	dFdC	dFdC
t_1/2_ (h)	9.91 ± 1.87	10.43 ± 0.91	10.56 ± 1.86
AUC_0–∞_ (ng·h/mL)	2691.3 ± 456.8	27,498.0 ± 5653.3	22,685 ± 4136.2
T_max_ (h)	2.00 ± 0.00	4.00 ± 0.00	0.083 ± 0.00
C_max_ (ng/mL)	451.1 ± 106.7	1656.3 ± 431.5	14,145 ± 4115

**Table 2 molecules-25-02218-t002:** Enzymatic kinetic parameters.

Parameters	Disappearance of Cbz-dFdC			Appearance of dFdC		
	V_max_ (nmol/min/mg pro)	K_m_ (μM)	Clint (mL/min/mg)	V_max_ (nmol/min/mg pro)	K_m_ (μM)	Clint (mL/min/mg)
Rat Blood	341.1	69.8	4.89	178.5	73.2	2.44
Rat Plasma	486.2	90.3	5.38	230.4	101.2	2.28
Rat Liver Homogenate	411.6	189.6	2.17	86.8	96.4	0.90
Rat Liver S9	169.6	96.3	1.76	49.5	41.2	1.20
Rat Kidney Homogenate	162.7	147.5	1.10	44.3	48.9	0.91
Rat Kidney S9	206.5	269.3	0.77	50.2	78.7	0.64
Human Liver S9	1491.3	107.8	13.83	480.6	65.7	7.32
Human CES1 Recombinase	18,802.3	258.6	72.71	3910.4	107.2	36.48

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
