# Peer review of "A Pharmacokinetic and Pharmacodynamic Evaluation of the Anti-Hepatocellular Carcinoma Compound 4-N-Carbobenzoxy-gemcitabine (Cbz-dFdC)"

_molecules, 2020, doi:10.3390/molecules25092218_

Round 1
Reviewer 1 Report
I analyzed the article ”A Pharmacokinetic and Pharmacodynamic Evaluation 3 of the Anti-hepatocellular Carcinoma Compound 4- 4 N-carbobenzoxy- gemcitabine (Cbz-dFdC)” and I found an improvement of the article. However, I still have to make some observations:
I think that it is not enough to attach the NMR spectra of Cbz-dFdC as an supplementary file, without being discussed to prove the chemical structure of the obtained compound.
In the supplementary material, the figures should be numbered (referring to them in the article), and they must be given a title.
The authors need to take care of some technical details such as break by point (line 37), space between numbers and units of measurement (line 185-187)
ml should be written mL
At References, pay attention to the writing of the year, which must be written in bold type
Considering the presented ones, I think that the article has to be further processed to reach the publication level.
Author Response
Dear Editors and Reviewers,
Thanks for your comments on our manuscript (applsci-760762, Title: A Pharmacokinetic and Pharmacodynamic Evaluation 3 of the Anti-hepatocellular Carcinoma Compound 4-4 N-carbobenzoxy- gemcitabine (Cbz-dFdC). Now, all changes were highlighted with blue in manuscript and responses to reviewers’ comments were included in this cover letter. We revised point to point according to instructions as follows:
Reviewer 1
I analyzed the article ”A Pharmacokinetic and Pharmacodynamic Evaluation 3 of the Anti-hepatocellular Carcinoma Compound 4- 4 N-carbobenzoxy- gemcitabine (Cbz-dFdC)” and I found an improvement of the article. However, I still have to make some observations:
I think that it is not enough to attach the NMR spectra of Cbz-dFdC as an supplementary file, without being discussed to prove the chemical structure of the obtained compound.
Answer: We have added some contents in discussion. 13C NMR (126 MHz, DMSO) δ 163.38 (C), 153.99 (C), 153.00 (C), 144.46 (CH), 135.81 (C), 128.48 (CH)2, 128.19 (CH), 127.96 (CH)2, 122.93 (t, JC−F = 259.6, C), 94.86 (CH), 84.08 (t, JC−F = 30.2, CH), 80.98 (CH), 68.37 (t, JC−F = 22.7, CH), 66.67 (CH2), 58.78 (CH2). 1H NMR (500 MHz,DMSO) δ 10.98 (s, 1H), 8.23 (d, J = 7.6, 1H), 7.45−7.32 (m, 5H), 7.11 (d, J = 7.6, 1H), 6.31 (d, J = 6.5, 1H), 6.16 (t, J = 7.4, 1H), 5.29 (t, J = 5.5, 1H), 5.20 (s, 2H), 4.25−4.13 (m, 1H), 3.88 (dt, J = 8.5, 3.1, 1H), 3.84−3.62 (m, 2H). MS (ESI) m/z 398.4 [M + H]+.
In the supplementary material, the figures should be numbered (referring to them in the article), and they must be given a title.
Answer: We have added titles and numbers in figures in supplementary materials.
The authors need to take care of some technical details such as break by point (line 37), space between numbers and units of measurement (line 185-187).ml should be written mL
Answer: We have revised these details in text.
At References, pay attention to the writing of the year, which must be written in bold type
Answer: We have revised these references in text.
Considering the presented ones, I think that the article has to be further processed to reach the publication level.
Reviewer 2 Report
This paper reports the PK and anti-tumor efficacy of a gemcitabine prodrug. Study itself seem to be interesting. However, the manuscript contains many unclear points. Specific points are detailed in my comments below.
- To show the better antitumor activity of the prodrug Cbz-dFdC compared to gemcitabine, antitumor efficacy study should include gemcitabine treated group.
- For pharmacokinetic study and in vivo efficacy study, it is unclear how many animals were evaluated in each group.
- I cannot find any statistical analysis results. For figure 8, t-test is not proper method for comparison.
- Figure 4, the concentrations of Cbz-dFdC and dFdC in the tissue homogenate were presented as ng/mg protein. Therefore, it was hard to compare this concentration with plasma concentrations. I recommend presenting tissue concentration as ng/g tissue.
- Enzyme kinetic study
- Page 4, line107-111; Formation of dFdC was not obvious in liver and kidney based on your supplemental data. If this might be due to faster conversion of dFdC to dFdU, it would be better to show dFdU generated. Have you also measured dFdU?
- Figure 5 and Table 2; Was the “V (nmol/min/mg protein)” value based on disappearance of Cbz-dFdC? This point should be clearly described.
- Page 4, line112; The sentence “The phenomenon was weaker~” is not understandable. What does “phenomenon” mean?
- Table 2; It would be better to add intrinsic clearance values (Vmax/Km) for comparison of hydrolysis activity of Cbz-dFdC in each tissue.
- Hydrolysis activity of Cbz-dFdC seem to be highest based on Table 2 and Figure 5. However, this not consistent with your supplemental data, which indicate the fastest degradation of Cbz-dFdC in rat plasma and blood.
- Line 169-170; Superior antitumor activity of Cbz-dFdC compared to gemcitabine is not supported by your results. Refer my comments 1.
- Line 172-174 & 178-187; I recommend comparing AUC of dFdC and AUC ratios of dFdU versus dFdC. In the last sentence, “The expected results” is not clear.
- Some sentences should be revised. For instance,
“This suggests that the accumulation of Cbz-dFdC in the intestine is not easily degraded.”
Author Response
Dear Editors and Reviewers,
Thanks for your comments on our manuscript (applsci-760762, Title: A Pharmacokinetic and Pharmacodynamic Evaluation 3 of the Anti-hepatocellular Carcinoma Compound 4-4 N-carbobenzoxy- gemcitabine (Cbz-dFdC). Now, all changes were highlighted with blue in manuscript and responses to reviewers’ comments were included in this cover letter. We revised point to point according to instructions as follows:
Reviewer2
This paper reports the PK and anti-tumor efficacy of a gemcitabine prodrug. Study itself seem to be interesting. However, the manuscript contains many unclear points. Specific points are detailed in my comments below.
To show the better antitumor activity of the prodrug Cbz-dFdC compared to gemcitabine, antitumor efficacy study should include gemcitabine treated group.
Answer: We added anti-tumor activity of dFdC and compared the anti-tumor activity between dFdC and CBZ-dFdC in mice (In Fig.8). At equimolar dose of dFdC and CBZ-dFdC , dFdC was treated by i.p. 15mg/kg once every 3 days for 4 times, while CBZ-dFdC was treated by i.g. 23mg/kg once every 3 days for 4 times. The results showed that the antitumor effects of CBZ-dFdC were slightly better than that of dFdC in vivo.
dFdC showed serious side effects on gastrointestinal tract (Int J Radiat Oncol Biol Phys 2012, 83, 1120–1125.), therefore it was injected intravenously in clinic. Compared with oral administration, intravenous administration increased psychological and physical burden in patients. Therefore, Cbz-dFdC, as a prodrug of dFdC, was designed for oral use. The gastrointestinal tract toxicity after oral administration of Cbz-dFdC was lower than that of oral administration of dFdC.
For pharmacokinetic study and in vivo efficacy study, it is unclear how many animals were evaluated in each group.
Answer: We added animals numbers in text.
I cannot find any statistical analysis results. For figure 8, t-test is not proper method for comparison.
Answer: We added statistical analysis results in text.
Figure 4, the concentrations of Cbz-dFdC and dFdC in the tissue homogenate were presented as ng/mg protein. Therefore, it was hard to compare this concentration with plasma concentrations. I recommend presenting tissue concentration as ng/g tissue.
Answer: We added the units of tissue concentrations in Figure 4.
Enzyme kinetic study
Page 4, line107-111; Formation of dFdC was not obvious in liver and kidney based on your supplemental data. If this might be due to faster conversion of dFdC to dFdU, it would be better to show dFdU generated. Have you also measured dFdU?
Answer: In enzyme kinetic study, we did not measure the concentration of dFdU. It is possible that dFdC in the incubation system can be metabolized to inactive dFdU by cytidine deaminase. However, our enzyme kinetics study mainly focused on the process of CBZ-dFdC to dFdC by CES1 esterase, so we measured CBZ-dFdC and dFdC in the present study. Additionally, we added the enzyme kinetic curve based on appearance of dFdC concentration in enzyme kinetic results.
Figure 5 and Table 2; Was the “V (nmol/min/mg protein)” value based on disappearance of Cbz-dFdC? This point should be clearly described.
Answer: In Figure 5 and Table 2, the “V (nmol/min/mg protein)” value was calculated based on disappearance of Cbz-dFdC. Additionally, we added the enzyme kinetic curve based on appearance of dFdC concentration in enzyme kinetic results.
Page 4, line112; The sentence “The phenomenon was weaker~” is not understandable. What does “phenomenon” mean?
Answer: We revised the sentences in text.
Table 2; It would be better to add intrinsic clearance values (Vmax/Km) for comparison of hydrolysis activity of Cbz-dFdC in each tissue.
Answer: We added the intrinsic clearance values in text.
Hydrolysis activity of Cbz-dFdC seem to be highest based on Table 2 and Figure 5. However, this not consistent with your supplemental data, which indicate the fastest degradation of Cbz-dFdC in rat plasma and blood.
Answer: According to the Table 2 and Figure 5, the intrinsic clearance values (Clint) in rat blood and plasma were larger than those in tissue homogenates and S9, which showed that Cbz-dFdC was easier to degrade in rat plasma and blood.
In supplemental materials, the initial concentration of CBZ-dFdC was 100ng/ml at time-dependent degraded curves. This concentration was much smaller than the minimum concentration( 0.5uM) in enzyme kinetic study, so the metabolized dFdC from CBZ-dFdC was very few at time-dependent degradation. That was why we deleted the time-dependent degradation results in supplemental material.
Line 169-170; Superior antitumor activity of Cbz-dFdC compared to gemcitabine is not supported by your results. Refer my comments 1.
Answer: We added antitumor activity of dFdC in results.
Line 172-174 & 178-187; I recommend comparing AUC of dFdC and AUC ratios of dFdU versus dFdC. In the last sentence, “The expected results” is not clear.
Answer: We added the pharmacokinetic curves of dFdC after a single i.v. dose in Figure 3. dFdC was degraded rapidly after intravenous injection in vivo. The initial plasma concentration of dFdC was high (about 15000 ng/ml) after an intravenous injection at the dose of 15mg/kg, but it decreased rapidly (about 130 ng/ml at 24h), the mean AUC0-∞ was 22685 ng·h/mL. Cbz-dFdC entered into the body and transformed into dFdC slowly after a single intragastric at the dose of 23 mg/kg. The peak time of dFdC after Cbz-dFdC administration was delayed to about 4h. The peak concentration at 4h was about 1500 ng/ml. The plasma concentration of dFdC remained 300 ng/ml at 24h and 130 ng/ml at 48h, and the mean AUC0-∞ was 27498.0 ng·h/mL, indicating that the drug had little fluctuation in vivo and remained for a long antitumor effect. Additionally, we also revised “The expected results”.
Some sentences should be revised. For instance,
“This suggests that the accumulation of Cbz-dFdC in the intestine is not easily degraded.”
Answer: We have revised some sentences in text.
Reviewer 3 Report
Manuscript entitled “A Pharmacokinetic and Pharmacodynamic Evaluation of the Anti-hepatocellular Carcinoma Compound 4-N-carbobenzoxy- gemcitabine (Cbz-dFdC)” describes a prodrug strategy to address the poor pharmacokinetics observed with dFdC. While the results describe an interesting approach that enables the sustained release of a drug with a short half-life, authors failed in getting the experimental design correct, as most of the experiments discussed here lack proper controls. I recommend major revisions prior to acceptance:
1) Poorly written introduction. There was no logical discussion about the possible benefit of Cbz-dFdC over LY2334737 or other 4-(N)-carbamate-modified dFdC prodrugs in structural perspective.
2) In the in vivo PK section, authors mentioned that “the formation of a considerable amount of dFdC and dFdU had an antitumor effect” (line 83). But in the introduction, they mentioned the dFdU to be an inactive metabolite. Please clarify or correct
3) For in vitro and in vivo experiments authors did not have control groups with dFdC alone treatment, which is essential to understand the benefits obtained from prodrug strategy. In the introduction authors claim significant antitumor effects of dFdC against solid tumors in vitro and invivo. dFdC release levels obtained after prodrug treatment vs dFdC alone treatment should be compared both in the plasma and tissues, more importantly at the target site of action and correlated with efficacy endpoints. Hence, it is recommended to have control group data to be generated under the similar experimental conditions.
4) What is the incubation concentration in stability experiments? Please include. BNPP incubations for matrices where it is clearly known that CES1 levels are insignificant is not needed. Please comment.
5) Line 122, Figure 6 do not represent a degradation profile, it demonstrates characterization of kinetics. Please correct.
6) It would be nice to see comparative data of Cbz-dFdC stability, and appearance of dFdC and dFdU across various matrices. Please provide a comprehensive figure.
7) For cell efficacy experiments, it is recommended to measure prodrug levels in the supernatant as well as cells. These observations help in understanding if the prodrug has lower capacity to permeate to cancer cells which could be one of the reasons why it has lower efficacy than the drug itself. Line 166-167, please revise the sentence “volume increased”, it is not clear.
8) If prodrug converts to drug in liver and plasma how did the authors believe that it has benefit it terms of achieving better efficacy. As soon as the prodrug hits the circulation it converts to drug in plasma and liver, suggesting the only benefit obtained here is the route of administration. It would be good to see the pharmacokinetic profiles of dFdC administered after dFdC administration. At places, authors compared the concentration levels of dFdC obtained after prodrug administration to published literature on dFdC. Did they take in to consideration the similarity in doses? Please comment.
9) The mechanism behind sustained release obtained from prodrug administration is not clearly discussed in the manuscript. What factors drive the sustained release behavior of prodrug? Please discuss.
10) Is PBS the formulation vehicle used for PK and efficacy studies? Please comment.
11) Line 205, which figure 5 are the authors referring to? Please correct.
12) Supplementary material do not have figure legends. Please mention.
13) Authors should use ANOVA for statistical analysis for in vivo efficacy study. Appropriate statistical parameters (p values) should always be mentioned where they used the term “significant”.
Author Response
Reviewer 3
Comments and Suggestions for Authors
Manuscript entitled “A Pharmacokinetic and Pharmacodynamic Evaluation of the Anti-hepatocellular Carcinoma Compound 4-N-carbobenzoxy- gemcitabine (Cbz-dFdC)” describes a prodrug strategy to address the poor pharmacokinetics observed with dFdC. While the results describe an interesting approach that enables the sustained release of a drug with a short half-life, authors failed in getting the experimental design correct, as most of the experiments discussed here lack proper controls. I recommend major revisions prior to acceptance:
Thanks for your comments on our manuscript (applsci-760762, Title: A Pharmacokinetic and Pharmacodynamic Evaluation of the Anti-hepatocellular Carcinoma Compound 4-4 N-carbobenzoxy- gemcitabine (Cbz-dFdC). Now, all changes were highlighted with red in manuscript and responses to reviewers’ comments were included in this cover letter. We revised point to point according to instructions as follows:
1) Poorly written introduction. There was no logical discussion about the possible benefit of Cbz-dFdC over LY2334737 or other 4-(N)-carbamate-modified dFdC prodrugs in structural perspective.
Answer: The acyl group in LY2334737 is more sterically hindered, which slows the hydrolysis of the prodrug and impedes a sustained release of gemcitabine in vivo. The stability of alkoxycarbonyl moieties is often better than that of amide structures. When applied to prodrug design, it is likely to exhibit more ideal pharmacokinetic properties and achieve slow release of gemcitabine in the body. We have added in introduction.
2) In the in vivo PK section, authors mentioned that “the formation of a considerable amount of dFdC and dFdU had an antitumor effect” (line 83). But in the introduction, they mentioned the dFdU to be an inactive metabolite. Please clarify or correct
Answer: We corrected the sentence in line 83.
3) For in vitro and in vivo experiments authors did not have control groups with dFdC alone treatment, which is essential to understand the benefits obtained from prodrug strategy. In the introduction authors claim significant antitumor effects of dFdC against solid tumors in vitro and invivo. dFdC release levels obtained after prodrug treatment vs dFdC alone treatment should be compared both in the plasma and tissues, more importantly at the target site of action and correlated with efficacy endpoints. Hence, it is recommended to have control group data to be generated under the similar experimental conditions.
Answer: Because dFdC was injected intravenously in clinic, we added the pharmacokinetic curves of dFdC after a single i.v. dose in Figure 3. dFdC was degraded rapidly after intravenous injection in vivo. The initial plasma concentration of dFdC was high (about 15000 ng/ml) after an intravenous injection at the dose of 15mg/kg, but it decreased rapidly (about 130 ng/ml at 24h), the mean AUC0-∞ was 22685 ng·h/mL. Cbz-dFdC entered into the body and transformed into dFdC slowly after a single intragastric at the dose of 22 mg/kg. The peak time of dFdC after Cbz-dFdC administration was delayed to about 4h. The peak concentration at 4h was about 1500 ng/ml. The plasma concentration of dFdC remained 300 ng/ml at 24h and 130 ng/ml at 48h, and the mean AUC0-∞ was 27498.0 ng·h/mL, indicating that the drug had little fluctuation in vivo and remained for a long antitumor effect.
We added anti-tumor activity of dFdC and compared the anti-tumor activity between dFdC and CBZ-dFdC in mice (In Fig.8). At equimolar dose of dFdC and CBZ-dFdC , dFdC was treated by i.p. 15mg/kg once every 3 days for 4 times, while CBZ-dFdC was treated by i.g. 23mg/kg once every 3 days for 4 times. The results showed that the antitumor effects of CBZ-dFdC were slightly better than that of dFdC in vivo.
dFdC showed serious side effects on gastrointestinal tract (Int J Radiat Oncol Biol Phys 2012, 83, 1120–1125.), therefore it was injected intravenously in clinic. Compared with oral administration, intravenous administration increased psychological and physical burden of patients. Therefore, Cbz-dFdC, as a prodrug of dFdC, was designed for oral use. The gastrointestinal tract toxicity after oral administration of Cbz-dFdC was lower than that of oral administration of dFdC.
4) What is the incubation concentration in stability experiments? Please include. BNPP incubations for matrices where it is clearly known that CES1 levels are insignificant is not needed. Please comment.
Answer: We have added the incubation concentration in stability experiments in text. In concentration-dependent incubation study, the drug incubation system was 5, 10, 20, 50, 100, 200 μM. It was true that BNPP can not exert the effect to inhibit CES1 activity when CES1 activity was relatively low, so BNPP can not be added.
5) Line 122, Figure 6 do not represent a degradation profile, it demonstrates characterization of kinetics. Please correct.
Answer: We have revised it in text.
6) It would be nice to see comparative data of Cbz-dFdC stability, and appearance of dFdC and dFdU across various matrices. Please provide a comprehensive figure.
Answer: In enzyme kinetic study, we did not measure the concentration of dFdU. It is possible that dFdC in the incubation system can be metabolized to inactive dFdU by cytidine deaminase. However, our enzyme kinetics study mainly focused on the process of CBZ-dFdC to dFdC by CES1 esterase, so we measured CBZ-dFdC and dFdC in the present study. Additionally, we added the enzyme kinetic curve based on appearance of dFdC concentration in enzyme kinetic results.
7) For cell efficacy experiments, it is recommended to measure prodrug levels in the supernatant as well as cells. These observations help in understanding if the prodrug has lower capacity to permeate to cancer cells which could be one of the reasons why it has lower efficacy than the drug itself. Line 166-167, please revise the sentence “volume increased”, it is not clear.
Answer: We measured the cell uptake of CBZ-dFdC and dFdC in HepG2 cells. The results showed that: 1) after addition of CBZ -dFdC, the uptake of CBZ-dFdC in HepG2 cells was very low in cell, and then hydrolyzed to dFdC in different concentrations, and the hydrolyzed dFdC concentrations increased with the increase of CBZ-dFdC concentration. 2) After dFdC treatment alone, the concentrations of dFdC in HepG2 cells also increased with the increase of dFdC concentration. 3) The intracellular concentration of dFdC after dFdC treatment alone was higher than that of dFdC after CBZ-dFdC treatment. These results may be one of the reasons that the anti-tumor effect of dFdC in HepG2 cells was better than that of Cbz-dFdC. However, considering that dFdC was released slowly after intragastric administration of Cbz-dFdC in vivo, Cbz-dFdC might be supposed to have some advantages in anti-tumor effect in vivo. Additionally, we also revised the sentence in Line 166-167.
8) If prodrug converts to drug in liver and plasma how did the authors believe that it has benefit it terms of achieving better efficacy. As soon as the prodrug hits the circulation it converts to drug in plasma and liver, suggesting the only benefit obtained here is the route of administration. It would be good to see the pharmacokinetic profiles of dFdC administered after dFdC administration. At places, authors compared the concentration levels of dFdC obtained after prodrug administration to published literature on dFdC. Did they take in to consideration the similarity in doses? Please comment.
Answer: At equimolar dose of dFdC and CBZ-dFdC, we compared the pharmacokinetic curves of dFdC after a single i.v. dose of dFdC at 15mg/kg with those of dFdC after a single i.g. dose of Cbz-dFdC at 23mg/kg in Figure 3. dFdC was degraded rapidly after intravenous injection in vivo. The initial plasma concentration of dFdC was high (about 15000 ng/ml) after an intravenous injection at the dose of 15mg/kg, but it decreased rapidly (about 130 ng/ml at 24h), the mean AUC0-∞ was 22685 ng·h/mL. Cbz-dFdC entered into the body and transformed into dFdC slowly after a single intragastric at the dose of 23 mg/kg. The peak time of dFdC after Cbz-dFdC administration was delayed to about 4h. The peak concentration at 4h was about 1500 ng/ml. The plasma concentration of dFdC remained 300 ng/ml at 24h and 130 ng/ml at 48h, and the mean AUC0-∞ was 27498.0 ng·h/mL, indicating that the drug had little fluctuation in vivo and remained for a long antitumor effect.
9) The mechanism behind sustained release obtained from prodrug administration is not clearly discussed in the manuscript. What factors drive the sustained release behavior of prodrug? Please discuss.
Answer: There were two main reasons for the sustained release of Cbz-dFdC. One was that Cbz-dFdC was given orally, so there was an absorption process, which gradually slowly entered the body to release dFdC. The other was that it took some time to metabolize into dFdC from Cbz-dFdC to enter the body, which also resulted in the slow-release of Cbz-dFdC.
10) Is PBS the formulation vehicle used for PK and efficacy studies? Please comment.
Answer: 0.9% normal saline was formulation vehicle used for PK and efficacy studies.
11) Line 205, which figure 5 are the authors referring to? Please correct.
Answer: We also revised these in text.
12) Supplementary material do not have figure legends. Please mention.
Answer: We also revised these in supplementary material.
13) Authors should use ANOVA for statistical analysis for in vivo efficacy study. Appropriate statistical parameters (p values) should always be mentioned where they used the term “significant”.
Answer: We also revised these in text.
Round 2
Reviewer 2 Report
The revised manuscript has been much improved. Most of comments were properly reflected in this version of manuscript.
Reviewer 3 Report
The comments are sufficiently addressed and I recommend acceptance of the revised manuscript.This manuscript is a resubmission of an earlier submission. The following is a list of the peer review reports and author responses from that submission.
Round 1
Reviewer 1 Report
In this manuscript, Sun and co-workers studied the pharmacokinetics and pharmacodynamics of Cbz-dFdC, a prodrug of gemcitabine, in rats. Although dFdC is believed to be effective, the usefulness is apparently limited by its inadequate pharmacokinetic properties. Therefore, the prodrug approach that prolongs the residence in the body may be kinetically relevant for this anti-cancer drug. The methodological approach appears straightforward and data interpretation is mostly adequate. However, there are some major and minor problems in the manuscript:
Major comments
- As pointed out by the authors, the rationale behind Cbz-dFdC is largely identical to that of LY2334737, another prodrug for dFdC, which is already in phase II study. Considering the fact that conceptually identical prodrug is already well into the development, what is the scientific novelty / practical justification of the current study? In addition, it is strongly recommended to re-do the experiment with LY2334737 so that the pharmacokinetic/pharmacodynamic performance of the prodrugs may be compared.
- Based on the data presented in Figs. 2-4, one can conclude that the prodrug is likely to be reasonably absorbed in the intestine, but is rapidly converted to the inactive form (dFdU). Although the pharmacokinetic constants were not presented to estimate other parameters, the mean residence time for the active form would be very short suggesting that the therapy may not be effective. Consistent with this statement, the treatment with high dose (22 mg/kg) for 15 consecutive days was required to produce a measurable effect. If this is the case, what is the scientific justification of the authors’ statement that ‘Cbz-dFdC is a good candidate for a potential prodrug’.
- As shown if Fig. 3, the residence of CBZ-dFdC in the body was brief and the plasma concentration low. Alternatively, the residence and the concentration of dFdC was slightly longer and higher. Therefore, it is recommended to do the experiment for Fig. 9 with dFdC to express the data of ‘Inhibition rate vs. dFdCconcentration’ since dFdC is likely to be the major form exerting the pharmacological activity.
Minor comments
- Although the data in Fig. 5 and 7 were relevant for the study, it is difficult to appreciate the importance because the individual figures were so small. It is recommended that the figure sizes are enlarged and presented in the supplementary section.
- Did the authors check the enzyme kinetics indeed follow Michaelis-Menten kinetics? Please explain whether standard transformation (e.g., Eadie-Hofstee plot) resulted in single component Michaelis-Menten kinetics.
Reviewer 2 Report
I have assesed the manuscript entitled “A Pharmacokinetic and Pharmacodynamic Evaluation 3 of the Anti-hepatocellular Carcinoma Compound 4- 4 N-carbobenzoxy- gemcitabine (Cbz-dFdC)” and I have the following observations:
Line 34-36 needs reformulation to avoid repetition
Line 167-168 chlorine formic acid benzyl ester must be written benzyl chloroformate
4.1. Materials and Equipment nothing is mentioned about the equipment used in NMR analysis
The article is interesting in terms of research to improve cancer treatment.
However, from a chemical point of view, it requires some improvements, such as:
- Performing 13C-NMR, and IR spectral analysis of Cbz-dFdC
- The NMR spectra should be attached as Supplementary Material
- Performing elementary analysis
- How was the purity of Cbz-dFdC determined?
Reviewer 3 Report
The pharmacodynamic results showed that the IC50 of Cbz-dFdC is about 0.12 μM in HepG2. The tumor inhibition of high and low doses of Cbz-dFdC in HepG2 tumor-bearing rats was 53% and 25%, respectively. In general, Cbz-dFdC has good pharmaceutical characteristics and is therefore a good candidate for a potential prodrug.
The article is interesting and can be useful in future applications. It deserves to be published, after correcting some forms of the English expression.